

# 2-Group symmetries in class S

**Lakshya Bhardwaj**

Mathematical Institute, University of Oxford,
Andrew Wiles Building, Woodstock Road, Oxford, OX2 6GG, UK

## Abstract

2-group symmetries are generalized symmetries that arise when 1-form and 0-form symmetries mix with each other. We uncover the existence of a class of 2-group symmetries in general $4d$ $\mathcal{N} = 2$ theories of Class S that can be constructed by compactifying $6d$ $\mathcal{N} = (2,0)$ SCFTs on Riemann surfaces carrying arbitrary regular punctures and outer-automorphism twist lines. The 2-group structure can be captured in terms of equivalence classes of line defects plus flavor Wilson lines, which can be thought of as accounting for screening of line defects while keeping track of flavor charges. We describe a method for computing these equivalence classes for a general Class S theory using the data on the Riemman surface used for compactifying its parent $6d$ $\mathcal{N} = (2,0)$ theory.

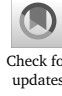
doi:[10.21468/SciPostPhys.12.5.152]()

# 1 Introduction

Generalized global symmetries have transformed our way of thinking about quantum field theory, bringing the study of extended defects to the fore. A systematic study of generalized symmetries was kicked off by [1] (see also related previous works [2–7]), where higher-form symmetries were defined. Higher-form symmetries of different types can mix together to give rise to generalized symmetries known as higher-group symmetries. In particular, 0-form and 1-form symmetries mix together to give rise to 2-group symmetries, which form the subject of this work.

2-group symmetries have been analyzed extensively in the recent literature [8–21]. However, such analyses have largely relied on using a Lagrangian description or are restricted to topological theories. In this paper, we analyze 2-group symmetries in $4d$ $\mathcal{N} = 2$ theories of Class S, which in general do not admit a conventional Lagrangian description.

For this analysis, we encode the 2-group symmetry in terms of properties of certain equivalence classes of line defects plus flavor Wilson lines. These equivalence classes are then read from the data of the compactification of the $6d$ $\mathcal{N} = (2,0)$ theory used to construct the Class S theory. The general idea of using compactifications of higher-dimensional theories to learn about generalized symmetries of lower-dimensional theories has proved to be very effective. See [21–39] for recent works employing this idea.

Let us now provide an overview of the paper. We begin in Section 2 by reviewing how 1-form symmetry groups can be understood in terms of equivalence classes of line defects with two line defects regarded in the same class if there exists a local operator living at their junction. We then point out that the computation of the global form of 0-form flavor symmetry groups can also be phrased in a similar language, where now instead of considering equivalence classes of line defects, one instead considers equivalence classes of flavor Wilson lines.

These considerations naturally lead to the idea that combining the two situations and considering equivalence classes of line defects plus flavor Wilson lines, which corresponds to keeping track of flavor center charges of junction local operators, should tell us something about the mixing of 1-form and 0-form symmetries. Indeed, as we discuss, these equivalence classes are related to a class of 2-group symmetries. In more detail, the three different equivalence classes discussed above fit into a short exact sequence whose associated Bockstein homomorphism controls the Postnikov class defining the 2-group symmetry [12].

In Section 3, we discuss these equivalence classes in the context of gauge theories. If the gauge group is simply connected, then one can straightforwardly compute the equivalence classes in terms of gauge and flavor center charges of the matter content of the gauge theory. For non-simply-connected gauge groups, the computation is more involved. See the soon to appear paper [40].

Section 4 discusses the computation of these equivalence classes for any arbitrary $4d$ $\mathcal{N} = 2$ Class S theory that can be obtained by compactifying a $6d$ $\mathcal{N} = (2,0)$ SCFT on a Riemann surface carrying regular twisted and untwisted punctures and possibly closed twist lines. The line defects of the $4d$ theory arise by compactifying surface defects of the $6d$ theory along 1-cycles on the Riemann surface. The junction local operators between $4d$ line defects are obtained by compactifying $6d$ surface defects along 2-chains whose boundary is formed by the 1-cycles corresponding to the $4d$ line defects. If the 2-chain passes over a puncture on the Riemann surface, the associated $4d$ junction local operator carries flavor center charges under the flavor algebra associated to the puncture. We describe a method of computing the flavor center charges of these junction local operators which generalizes the method described in [32] for computing the flavor center charges of genuine (non-junction) local operators. In the process, we also formulate the analysis of [31], for computing 1-form symmetries of Class

S theories, in more invariant terms that does not require a choice of A and B cycles on the punctured Riemann surface.

In Section 5, we illustrate the general procedure of Section 4 by implementing it in detail for a particular Class S theory. This theory also admits a Lagrangian description as $4d$ $\mathcal{N} = 2$ gauge theory with $\mathfrak{so}(4n+2)$ gauge algebra and $4n$ hypers in vector representation of the gauge algebra. The method of Section 4 predicts a non-trivial 2-group symmetry for this theory if the gauge group is $Spin(4n + 2)$, which can be verified by applying the analysis of Section 3. This provides a check of the general prescription of Section 4.

Generalizing the considerations of this paper to include irregular punctures would be an interesting future direction.

## 2 Generalities

### 2.1 1-Form Symmetries

The genuine line defects[1] of a QFT $\mathfrak{T}$ can be classified by imposing an equivalence relation[2] which regards two line defects $L_1$ and $L_2$ to be equivalent if there exists a non-zero local operator living at the junction of $L_1$ and $L_2$. Let us furthermore say that we do not keep account of the the flavor charges of junction local operators implementing the equivalences. Let the resulting set of equivalence classes be denoted as $\widehat{\mathcal{O}}$. In this paper, we only study those QFTs for which $\widehat{\mathcal{O}}$ forms a finite abelian group under the OPE of line defects.

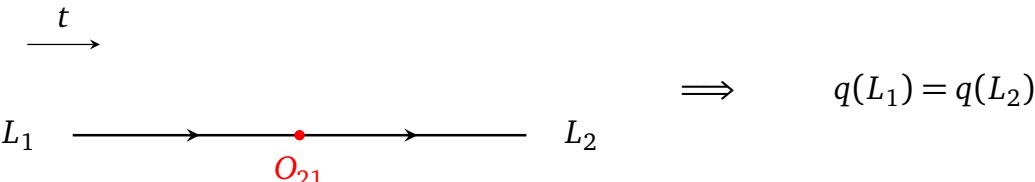

Figure 1: Consider two line defects $L_1$ and $L_2$ such that there exists a junction local operator $O_{21} \neq 0$ between them. Then, we can construct a configuration as shown on the left side of figure with time flowing horizontally. For charge conservation, the charges $q(L_1)$ and $q(L_2)$ of $L_1$ and $L_2$ must match.

The relevance of the above equivalence relation becomes manifest as one tries to define topological operators measuring charges of line defects. Then, $L_1$ and $L_2$ must have same charge if there is a local operator living at their junction. See Figure 1. Thus the set of possible charges is the abelian group $\widehat{\mathcal{O}}$, and the topological operators are specified by elements of the Pontryagin dual $\mathcal{O}$ of $\widehat{\mathcal{O}}$. In other words, $\mathcal{O}$ is the 1-form symmetry group of $\mathfrak{T}$. The theory $\mathfrak{T}$ can be coupled to these topological operators by turning on a background $[B_2] \in H^2(M, \mathcal{O})$ (where $M$ is a compact spacetime manifold) for the 1-form symmetry, which places the topological operators at the location of the Poincare dual of $[B_2]$.

### 2.2 Global Form of 0-Form Flavor Symmetry Group

Now we consider flavor Wilson lines in a similar language. Let

$$\mathfrak{f} = \mathfrak{f}_{na} \oplus \mathfrak{u}(1)^a \tag{1}$$

---

[1]A non-genuine line defect is one that is constrained to live at the boundaries or corners of higher-dimensional defects. On the other hand, a genuine line defect exists independently of any higher-dimensional defects.

[2]This equivalence relation is also known as 'screening' in the literature.

be the flavor symmetry algebra of $\mathfrak{T}$, such that $\mathfrak{f}_{na}$ is a non-abelian semi-simple Lie algebra and $\mathfrak{u}(1)^a$ is the abelian part of $\mathfrak{f}$. Now let us define

$$F = F_{na} \times U(1)^a \tag{2}$$

to be a group whose associated Lie algebra is $\mathfrak{f}$, such that $F_{na}$ is the simply connected group associated to $\mathfrak{f}_{na}$ and $U(1)^a$ is a group[3] whose associated Lie algebra is $\mathfrak{u}(1)^a$. Let us consider flavor Wilson lines transforming in representations of $F$ and impose an equivalence relation which regards two flavor Wilson lines $R_1$ and $R_2$ (where $R_1$ and $R_2$ are two representations of $F$) to be equivalent if there exists a *genuine* local operator[4] living at the junction of $R_1$ and $R_2$. Thus, $R_1$ is equivalent to $R_2$ if $\mathfrak{T}$ contains a genuine local operator transforming in representation $R_2 \otimes R_1^*$ where $R_1^*$ is the complex conjugate of $R_1$. Let the resulting set of equivalence classes of flavor Wilson lines be denoted as $\widehat{\mathcal{Z}}$.

As discussed in [32], the existence of current local operator for the flavor symmetry ensures that $\widehat{\mathcal{Z}}$ can be described as the abelian group

$$\widehat{\mathcal{Z}} = \widehat{Z}_F / Y_F \,, \tag{3}$$

where $\widehat{Z}_F$ is the Pontryagin dual of the center $Z_F$ of $F$ (which can be written as $Z_F = Z_{F,na} \times U(1)^a$ where $Z_{F,na}$ is the center of the simply connected group $F_{na}$) and $Y_F$ is the subgroup of $\widehat{Z}_F$ formed by the flavor center charges of genuine local operators of $\mathfrak{T}$.

The Pontryagin dual $\mathcal{Z}$ of $\widehat{\mathcal{Z}}$ is a subgroup of $Z_F$ and is instrumental in specifying the 0-form flavor symmetry group $\mathcal{F}$ of $\mathfrak{T}$, which can be written as

$$\mathcal{F} = F/\mathcal{Z} \,. \tag{4}$$

The theory $\mathfrak{T}$ can be coupled to a background principal bundle for $\mathcal{F}$, which is specified by a map

$$B_1 : \ M \to B\mathcal{F} \,, \tag{5}$$

where $B\mathcal{F}$ is the classifying space of $\mathcal{F}$. Such a bundle cannot always be lifted to an $F$ bundle. The obstruction to lifting is captured by the pullback $B_1^*[w_2] \in H^2(M, \mathcal{Z})$ of a characteristic class

$$[w_2] \in H^2(B\mathcal{F}, \mathcal{Z}) \,. \tag{6}$$

Let $B_1^* w_2 \in C^2(M, \mathcal{Z})$ be a $\mathcal{Z}$-valued 2-cochain on $M$ which is a representative of $B_1^*[w_2]$. Then we can think of the locus specified by Poincare dual of $B_1^* w_2$ as location of "topological operators valued in $\mathcal{Z}$ under which the flavor Wilson lines valued in $\widehat{\mathcal{Z}}$ are charged" since moving a part of this locus across a flavor Wilson line produces a phase factor determined in terms of the pairing $\widehat{\mathcal{Z}} \times \mathcal{Z} \to U(1)$.

Thus, the relationship between flavor Wilson lines in $\widehat{\mathcal{Z}}$ and background field $B_1^* w_2$ is the same as the relationship between line defects in $\widehat{\mathcal{O}}$ and background field $B_2$.[5]

## 2.3 2-Group Symmetries

Now consider the set of line defects plus flavor Wilson lines, which can be described by elements of the form $(L, R)$ where $L$ is a line defect and $R$ is a representation of $F$. Consider

---

[3]We would furthermore require that the charges of local operators under each $U(1)$ factor in $U(1)^a$ are integers.

[4]A non-genuine local operator is one that is constrained to live at the boundaries or corners of higher-dimensional defects. On the other hand, a genuine local operator exists independently of any higher-dimensional defects.

[5]In fact, this is one way to understand why turning on a 1-form symmetry background for a non-abelian gauge theory corresponds to turning on non-trivial $w_2$ for the gauge bundle.

imposing the same equivalence relation as in the last two subsections. That is, if there exists a local operator living at the junction of $L_1$ and $L_2$ transforming in representation $R_2 \otimes R_1^*$, then declare $(L_1, R_1)$ to be equivalent to $(L_2, R_2)$. Let the resulting set of equivalence classes be denoted as $\widehat{\mathcal{E}}$. Then, the three groups discussed above form a short exact sequence

$$0 \to \widehat{\mathcal{Z}} \to \widehat{\mathcal{E}} \to \widehat{\mathcal{O}} \to 0 \,, \tag{7}$$

because flavor Wilson lines (without any attached line defects) in $\widehat{\mathcal{Z}}$ form a subgroup of $\widehat{\mathcal{E}}$ and modding out the data of flavor Wilson lines from $\widehat{\mathcal{E}}$ should result in $\widehat{\mathcal{O}}$, by definition.

Let $B_w \in C^2(M, \mathcal{E})$ be the background field whose Pontryagin dual describes the location of topological operators labeled by $\mathcal{E}$ under which the lines in $\widehat{\mathcal{E}}$ are charged. The three background fields $B_2, B_w, B_1^* w_2$ are related to each other via the short exact sequence

$$0 \to C^2(M, \mathcal{O}) \to C^2(M, \mathcal{E}) \to C^2(M, \mathcal{Z}) \to 0 \,, \tag{8}$$

induced by the short exact sequence

$$0 \to \mathcal{O} \to \mathcal{E} \to \mathcal{Z} \to 0 \,. \tag{9}$$

Pontryagin dual to the short exact sequence (7). We have

$$B_w = \widetilde{w}_2 + i(B_2) \,, \tag{10}$$

where $\widetilde{w}_2 \in C^2(M, \mathcal{E})$ is a lift of $B_1^* w_2 \in C^2(M, \mathcal{Z})$ under the projection map

$$C^2(M, \mathcal{E}) \to C^2(M, \mathcal{Z}) \tag{11}$$

in (8) and $i(B_2) \in C^2(M, \mathcal{E})$ is the image of the 1-form symmetry background $B_2 \in C^2(M, \mathcal{O})$ under the injection map

$$i : \quad C^2(M, \mathcal{O}) \to C^2(M, \mathcal{E}) \tag{12}$$

in (8).

A consequence is that the 1-form symmetry background $B_2$ is not necessarily closed. Acting with differential on (10), we obtain

$$\delta \widetilde{w}_2 + i(\delta B_2) = 0 \,, \tag{13}$$

since $B_w$ is closed. This can be further rewritten as

$$i\left(B_1^* w_3 + \delta B_2\right) = 0 \,, \tag{14}$$

where $w_3 \in C^3(B\mathcal{F}, \mathcal{O})$ is a representative of the class

$$\text{Bock}[w_2] \in H^3(B\mathcal{F}, \mathcal{O}) \,, \tag{15}$$

obtained by applying to $[w_2]$ the Bockstein homomorphism

$$\text{Bock} : \quad H^2(B\mathcal{F}, \mathcal{Z}) \to H^3(B\mathcal{F}, \mathcal{O}) \,, \tag{16}$$

associated to the short exact sequence (9). Since $i$ is an injection, we deduce that

$$\delta B_2 + B_1^* w_3 = 0 \,, \tag{17}$$

which means that the 1-form symmetry background $B_2$ is non-closed.

In general, if the 1-form and 0-form symmetry backgrounds of a theory satisfy the relation

$$\delta B_2 + B_1^* \Theta = 0 \,, \tag{18}$$

such that $0 \neq [\Theta] \in H^3(B\mathcal{F}, \mathcal{O})$, then we say that the 1-form and 0-form symmetries of the theory mix with each other to form a non-trivial 2-group symmetry with *Postnikov class* $[\Theta]$. It should be noted that there is no 2-group symmetry if $[\Theta] = 0$ since then we can pick a representative $\Theta = 0 \in C^3(B\mathcal{F}, \mathcal{O})$ which restores $\delta B_2 = 0$. Notice that another consequence of a non-trivial 2-form symmetry is that

$$B_1^*[\Theta] = 0 \,, \tag{19}$$

which constrains the possible 0-form symmetry backgrounds $B_1$.

Thus, we learn that if line defects and flavor Wilson lines of a theory $\mathfrak{T}$ mix with each other as in (7), then we have a *potential* 2-group symmetry with Postnikov class

$$[\Theta] = \text{Bock}[w_2] \,, \tag{20}$$

where $[w_2] \in H^2(B\mathcal{F}, \mathcal{Z})$ is the characteristic class capturing the obstruction of lifting bundles for the 0-form flavor symmetry group $\mathcal{F} = F/\mathcal{Z}$ to bundles for the group $F$ discussed above, and Bock is the Bockstein homomorphism associated to (9). The 2-group symmetry degenerates to a direct product of 1-form symmetry $\mathcal{O}$ and 0-form symmetry $\mathcal{F}$ whenever the Postnikov class (20) vanishes. In such a situation, we say that the 2-group symmetry is *trivial*, which is a slight abuse of language since $\mathcal{O}$ and $\mathcal{F}$ can still be non-trivial.

An example of such a situation occurs if the short exact sequence (9) splits, which is equivalent to the splitness of the short exact sequence (7). Splitness means that we can write $\mathcal{E}$ as a direct product of the form

$$\mathcal{E} \simeq i(\mathcal{O}) \times \mathcal{Y} \,, \tag{21}$$

where $i(\mathcal{O}) \subseteq \mathcal{E}$ is the image of the map $\mathcal{O} \to \mathcal{E}$, and $\mathcal{Y}$ is some subgroup of $\mathcal{E}$. When a short exact sequence splits, the associated Bockstein homomorphism becomes the trivial homomorphism and hence the Postnikov class $[\Theta]$ in (20) vanishes.

## 3 Gauge Theories

In this section, we study the various generalized symmetries discussed in previous section for gauge theories. Consider a $4d$ gauge theory with a gauge algebra

$$\mathfrak{g} = \bigoplus_i \mathfrak{g}_i \,, \tag{22}$$

such that each $\mathfrak{g}_i$ is a non-abelian finite simple Lie algebra. Furthermore, assume that the gauge group of the gauge theory is

$$G = \prod_i G_i \,, \tag{23}$$

such that each $G_i$ is simply connected group associated to the Lie algebra $\mathfrak{g}_i$. Let $\mathfrak{f}$ be the flavor algebra of the gauge theory and let $F$ be the group associated to the flavor algebra as in (2).

The gauge theory carries Wilson-'t Hooft line defects and flavor Wilson lines. Before accounting for the matter content, in any gauge theory of the above type, we can restrict the possible charges of lines to lie in

$$\widehat{Z}_G \times \widehat{Z}_F \,, \tag{24}$$

where $\widehat{Z}_G$ is the Pontryagin dual of the center $Z_G$ of the gauge group $G$, which can be written as $Z_G = \prod_i Z_i$ where $Z_i$ is the center of $G_i$. The elements of $\widehat{Z}_G$ in (24) are purely Wilson line defects and the elements of $\widehat{Z}_F$ in (24) are flavor Wilson lines.

Now, the charges of matter content of the gauge theory form a sublattice

$$\mathcal{M} \subseteq \widehat{Z}_G \times \widehat{Z}_F. \tag{25}$$

These are the charges of genuine and non-genuine local operators arising due to the presence of the matter content. The charges of purely genuine local operators form a subgroup

$$Y_F = \mathcal{M} \cap \widehat{Z}_F \tag{26}$$

of $\mathcal{M}$, which captures the flavor center charges of gauge-invariant local operators of the gauge theory. Moreover, forgetting the flavor charges, the matter content gives rise to charges lying in a subgroup $Y_G \subseteq \widehat{Z}_G$ which can be described as

$$Y_G = \pi(\mathcal{M}), \tag{27}$$

if $\pi$ denotes the projection map $\widehat{Z}_G \times \widehat{Z}_F \to \widehat{Z}_G$.

The various groups discussed so far sit in a matrix of short exact sequences

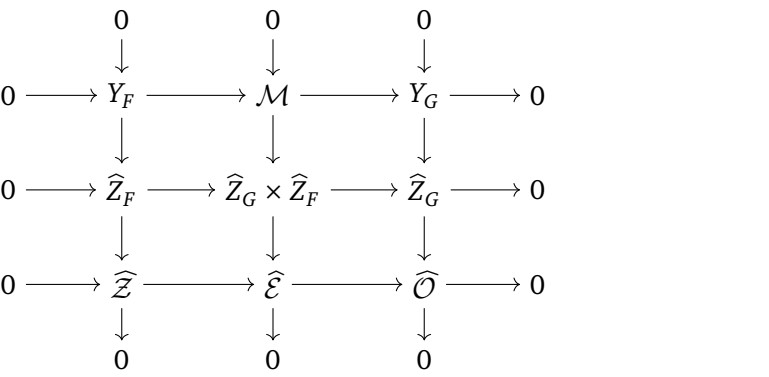

$$\tag{28}$$

such that every row and column forms a short exact sequence and every square commutes. The short exact sequence (7) responsible for potential 2-group symmetry (20) appears in the last row of (28).

In other words, $\mathcal{O}, \mathcal{E}, \mathcal{Z}$ are subgroups respectively of $Z_G, Z_G \times Z_F, Z_F$ under which the elements of $Y_G, \mathcal{M}, Y_F$ respectively are uncharged. Moreover, $\mathcal{O}$ is a subgroup of $\mathcal{E}$ such that

$$\mathcal{O} = \mathcal{E} \cap Z_G, \tag{29}$$

and $\mathcal{Z}$ is obtained from $\mathcal{E}$ by forgetting information about $Z_G$, i.e.

$$\mathcal{Z} = \pi(\mathcal{E}), \tag{30}$$

if $\pi$ denotes the projection map $Z_G \times Z_F \to Z_F$. The flavor symmetry group $\mathcal{F}$ of the gauge theory is then as described as in (4), which captures the fact that the elements of $\mathcal{Z} \subset F$ are equivalent to gauge transformations specified by the projection of $\mathcal{E}$ onto $Z_G$, and so should not be regard as flavor symmetries of the theory. In fact, the groups $\mathcal{O}, \mathcal{E}, \mathcal{Z}$ appeared in this form in the recent work [21].

The above considerations are modified if the gauge group is chosen to be

$$G' = G/\Gamma, \tag{31}$$

with $\Gamma \subseteq Z_G$, where in general we now also have to choose a discrete theta parameter [3]. Now, one needs to take into account the flavor center charges of local operators living at the ends of magnetic and dyonic Wilson-'t Hooft line defects, which receive extra contributions due to fermions in the matter content. A classic example is that in a $4d$ $\mathcal{N} = 2$ theory with $SU(2)$ gauge group and $N_f$ hypers in fundamental, the monopole acquires flavor charges and transforms in a spinor irrep of $\mathfrak{so}(2N_f)$ flavor symmetry algebra [41]. We do not perform a systematic analysis of potential 2-group symmetries for such gauge theories in this paper, but see the soon to appear paper [40] for more details. It should be noted, however, that these extra contributions do not impact the calculation of $\mathcal{Z}$, which can still be described as the subgroup $Z_F$ under which the elements of $Y_F$ are uncharged, where

$$Y_F = \mathcal{M} \cap \widehat{Z}_F \,, \tag{32}$$

and $\mathcal{M} \subseteq \widehat{Z}_{G'} \times \widehat{Z}_F$ captures the charges of the matter content of the gauge theory. In other words, the flavor symmetry group

$$\mathcal{F} = F/\mathcal{Z} \tag{33}$$

is left unchanged as one changes the global form of the gauge group and/or discrete theta parameters.

# 4 Class S

In this section, we describe a general method for obtaining the three main groups $\widehat{\mathcal{O}}$, $\widehat{\mathcal{Z}}$ and $\widehat{\mathcal{E}}$, and the short exact sequence (7) associated to them for $4d$ $\mathcal{N} = 2$ theories of Class S. As discussed in Section 2, this short exact sequence captures a 2-group symmetry of the theory.

## 4.1 $6d$ $\mathcal{N} = (2, 0)$ Theories and Their Surface Defects

We begin by reviewing some facts about dimension-2 surface defects in $6d$ $\mathcal{N} = (2, 0)$ theories. Consider a $6d$ $\mathcal{N} = (2, 0)$ theory which is specified by a finite simple Lie algebra $\mathfrak{g}$ of $A, D, E$ type. Let $\mathcal{G}$ be the simply connected group associated to $\mathfrak{g}$. Dimension-2 surface defects of the $\mathcal{N} = (2, 0)$ theory play an important role in our considerations. Akin to the discussion of line operators above, we can classify these surface defects by imposing an equivalence relation which regards two surface defects $S_1$ and $S_2$ to be equivalent if there exists a non-zero line defect living at the junction of $S_1$ and $S_2$. The resulting set of equivalence classes forms an abelian group $\widehat{Z}_{\mathcal{G}}$ which is the Pontryagin dual of the center $Z_{\mathcal{G}}$ of the group $\mathcal{G}$.

These $\mathcal{N} = (2, 0)$ theories are non-genuine or relative $6d$ theories, that is they live at the boundaries of $7d$ topological QFTs. The dimension-2 surface defects of the $6d$ theory arise via topological dimension-3 hyper-surface defects of the $7d$ theory ending at the $6d$ boundary. A consequence of this is that taking a surface defect $S_1$ around another surface defect $S_2$ on the $6d$ boundary produces a braiding between the corresponding hyper-surface defects in the $7d$ bulk, and resolving this braiding produces a phase in the correlation function, implying that the surface defects of the $6d$ theory are mutually non-local. This phase can be expressed as

$$\exp\bigl(2\pi i \langle \alpha, \beta \rangle\bigr), \tag{34}$$

where $\alpha, \beta \in \widehat{Z}_{\mathcal{G}}$ are the equivalence classes that $S_1$ and $S_2$ live in respectively. That is, the non-locality of surface defects of the $6d$ theory is captured in a bi-homomorphism

$$\langle \cdot, \cdot \rangle : \ \widehat{Z}_{\mathcal{G}} \times \widehat{Z}_{\mathcal{G}} \to \mathbb{R}/\mathbb{Z}\,, \tag{35}$$

which is often referred to as *pairing*.

An $\mathcal{N} = (2,0)$ theory of type $\mathfrak{g}$ has a discrete 0-form symmetry group isomorphic to the group $\mathcal{O}_{\mathfrak{g}}$ of outer automorphisms of the finite simple Lie algebra $\mathfrak{g}$. The 0-form symmetry acts on the surface defects as outer-automorphisms act on $\widehat{Z}_{\mathcal{G}}$.

Let us collect all this data for reference:

- For $\mathfrak{g} = \mathfrak{su}(n)$, we have $\widehat{Z}_{\mathcal{G}} \simeq \mathbb{Z}_n$, and the pairing is specified by

$$\langle f, f \rangle = \frac{1}{n}, \tag{36}$$

  where $f$ is the center charge of the fundamental irrep of $\mathfrak{g} = \mathfrak{su}(n)$. This determines the pairing on $\widehat{Z}_{\mathcal{G}} \simeq \mathbb{Z}_n$ as $f$ is its generator.
  We have $\mathcal{O}_{\mathfrak{g}} \simeq \mathbb{Z}_2$ that acts by sending an element of $\widehat{Z}_{\mathcal{G}} \simeq \mathbb{Z}_n$ to its inverse.

- For $\mathfrak{g} = \mathfrak{so}(4n)$, we have $\widehat{Z}_{\mathcal{G}} \simeq \mathbb{Z}_2^s \times \mathbb{Z}_2^c$. The center charges $s$ and $c$ associated to the spinor and cospinor irreps of $\mathfrak{g} = \mathfrak{so}(4n)$ generate $\mathbb{Z}_2^s$ and $\mathbb{Z}_2^c$ respectively.
  For $n > 2$, we have $\mathcal{O}_{\mathfrak{g}} \simeq \mathbb{Z}_2$ that acts by interchanging $s$ and $c$.
  For $n = 2$, we have $\mathcal{O}_{\mathfrak{g}} \simeq S_3$, i.e. the permutation group of three objects. It acts by permuting $s$, $c$ and $v = s + c$.
  For $n = 2m$, the pairing is

$$\langle s, s \rangle = 0, \quad \langle c, c \rangle = 0, \quad \langle s, c \rangle = \frac{1}{2}. \tag{37}$$

  For $n = 2m + 1$, the pairing is

$$\langle s, s \rangle = \frac{1}{2}, \quad \langle c, c \rangle = \frac{1}{2}, \quad \langle s, c \rangle = 0. \tag{38}$$

- For $\mathfrak{g} = \mathfrak{so}(4n + 2)$, we have $\widehat{Z}_{\mathcal{G}} \simeq \mathbb{Z}_4$. The center charge $s$ associated to the spinor irrep of $\mathfrak{g} = \mathfrak{so}(4n + 2)$ generates $\widehat{Z}_{\mathcal{G}} \simeq \mathbb{Z}_4$.
  We have $\mathcal{O}_{\mathfrak{g}} \simeq \mathbb{Z}_2$ that acts by sending an element of $\widehat{Z}_{\mathcal{G}} \simeq \mathbb{Z}_4$ to its inverse.
  For $n = 2m$, the pairing is

$$\langle s, s \rangle = \frac{3}{4}. \tag{39}$$

  For $n = 2m + 1$, the pairing is

$$\langle s, s \rangle = \frac{1}{4}. \tag{40}$$

- For $\mathfrak{g} = \mathfrak{e}_6$, we have $\widehat{Z}_{\mathcal{G}} \simeq \mathbb{Z}_3$. The center charge $f$ associated to the **27** dimensional irrep of $\mathfrak{g} = \mathfrak{e}_6$ generates $\widehat{Z}_{\mathcal{G}} \simeq \mathbb{Z}_3$.
  We have $\mathcal{O}_{\mathfrak{g}} \simeq \mathbb{Z}_2$ that acts by sending an element of $\widehat{Z}_{\mathcal{G}} \simeq \mathbb{Z}_3$ to its inverse.
  The pairing is

$$\langle f, f \rangle = \frac{2}{3}. \tag{41}$$

- For $\mathfrak{g} = \mathfrak{e}_7$, we have $\widehat{Z}_{\mathcal{G}} \simeq \mathbb{Z}_2$. The center charge $f$ associated to the **56** dimensional irrep of $\mathfrak{g} = \mathfrak{e}_7$ generates $\widehat{Z}_{\mathcal{G}} \simeq \mathbb{Z}_2$.
  The outer-automorphism group is trivial.
  The pairing is

$$\langle f, f \rangle = \frac{1}{2}. \tag{42}$$

- For $\mathfrak{g} = \mathfrak{e}_8$, we have $\widehat{Z}_{\mathcal{G}} = 0$. The outer-automorphism group is trivial.

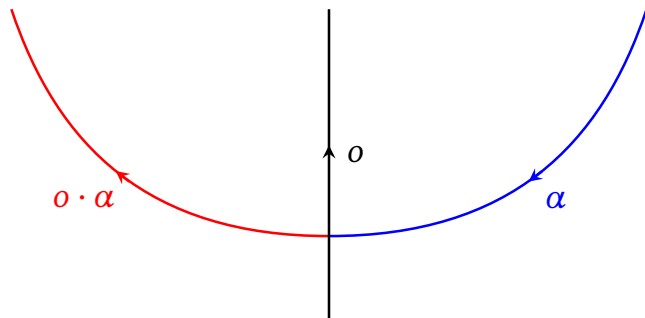

Figure 2: An outer-automorphism twist line (shown in black) acts on the element of $\widehat{Z}_{\mathcal{G}}$ carried by a K1-chain (shown in red and blue).

## 4.2 1-Form Symmetries

Consider compactifying a 6d $\mathcal{N} = (2,0)$ theory of type $\mathfrak{g} = A, D, E$ on a Riemann surface $\mathcal{C}$ of genus $g$. We allow $\mathcal{C}$ to carry untwisted and twisted *regular* punctures, and also closed twist lines. The resulting 4d $\mathcal{N} = 2$ Class S theory is also a non-genuine, relative theory that lives at the boundary of a 5d TQFT. This is reflected in the fact that the line defects of the 4d theory are mutually non-local as they live at the ends of dimension-2 topological surface defects of the 5d theory.

The line defects of the 4d theory can be obtained by compactifying dimension-2 surface defects of the 6d theory along 1-cycles of $\mathcal{C}$. We are interested in only keeping track of 4d line defects modulo screenings a.k.a. junction local operators, as above. Thus, we can restrict the surface defects of the 6d theory to lie in $\widehat{Z}_{\mathcal{G}}$. To see this, consider two surface defects $S_1$ and $S_2$ belonging to the same class in $\widehat{Z}_{\mathcal{G}}$ and let their compactification on some cycle $C$ give rise to two line defects $L_1$ and $L_2$ in the 4d theory. Then, there exists a non-zero local operator in the 4d theory living at the junction of $L_1$ and $L_2$, which is obtained by wrapping on $C$ the line defect in the 6d theory living at the junction of $S_1$ and $S_2$. This local operator between $L_1$ and $L_2$ does not carry any flavor charges.

Thus, we can restrict our attention to line defects obtained by *consistent* wrappings of 6d surface defects valued in $\widehat{Z}_{\mathcal{G}}$ along 1-cycles of the punctured Riemann surface $\mathcal{C}$. Since $\mathcal{C}$ carries outer-automorphism twist lines, which act on elements of $\widehat{Z}_{\mathcal{G}}$, it requires a bit of effort to define what a "consistent wrapping" means. To make this precise, we define the notions of K1 and K2 chains, cycles and homologies.

Consider a 1-chain $C$ that crosses an $o$-twist line. Let $C$ carry an element $\alpha \in \widehat{Z}_{\mathcal{G}}$ on one side of an $o$-twist line. Then, a consistent wrapping requires $C$ to carry the element $o \cdot \alpha \in \widehat{Z}_{\mathcal{G}}$ on the other side of the $o$-twist line. See Figure 2. Let us call such a consistent wrapping along a 1-chain as a *K1-chain*. Similarly, a *K1-cycle* is a consistent wrapping of $\widehat{Z}_{\mathcal{G}}$ surface defects along a 1-cycle on $\mathcal{C}$. In particular, the element of $\widehat{Z}_{\mathcal{G}}$ carried by a K1-cycle returns back to itself under the action of all the outer-automorphisms associated to all the outer-automorphism twist lines that the associated 1-cycle crosses. Note that we require that a puncture cannot lie inside the underlying 1-chains and 1-cycles associated to K1-chains and K1-cycles.

In a similar fashion, we define K2-chains, which have the property that if $p$ and $p'$ are two points lying on the underlying 2-chain such that there exists a path $\lambda$ connecting $p$ and $p'$ that crosses an $o$-twist line, and the 2-chain carries the element $\alpha \in \widehat{Z}_{\mathcal{G}}$ at $p$, then the element carried by the 2-chain at $p'$ must be $o \cdot \alpha \in \widehat{Z}_{\mathcal{G}}$. This provides constraints on $\alpha$ as there might exist two paths $\lambda_1$ and $\lambda_2$ both connecting $p$ and $p'$, but such that $\lambda_1$ crosses $o_1$-twist line and $\lambda_2$ crosses $o_2$-twist line with $o_1 \neq o_2$, then we must have $o_1 \cdot \alpha = o_2 \cdot \alpha$. K2-chains describe consistent compactifications of surface defects over the corresponding 2-chains. K2-chains can be of two types: non-extended and extended. A *non-extended* K2-chain is one whose

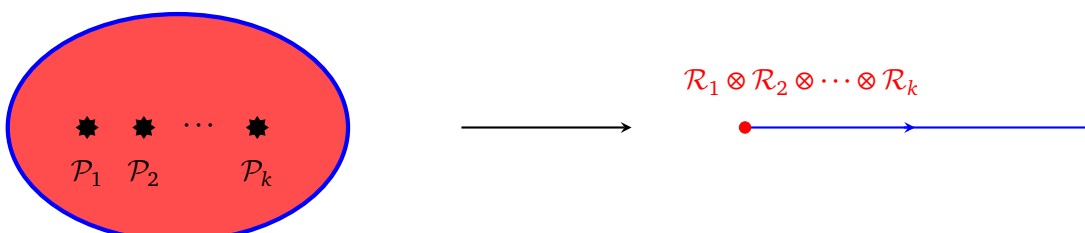

Figure 3: A 6$d$ surface defect compactified on a K2-chain (shown in red) whose boundary is a K1-cycle (shown in blue) leads to a 4$d$ local operator (shown in red) living at the end of a line defect corresponding to the K1-cycle (shown in blue). If the K2-chain passes over a puncture $\mathcal{P}_i$, then the local operator is charged under a representation $\mathcal{R}_i$ of $\mathcal{P}_i$. See (60).

underlying 2-chain does not contain any puncture. On the other hand, an *extended* K2-chain is one whose underlying 2-chain contains atleast one puncture.

The boundary of a K2-chain is a K1-cycle. So, we can define two kinds of K1-homologies: extended K1-homology $\mathcal{L}$ and non-extended K1-homology $\mathcal{K}$, depending on whether we regard or do not regard K1-cycles living at the boundaries of extended K2-chains as trivial.

These homologies carry a pairing bi-homomorphism $\langle \cdot, \cdot \rangle_{\mathcal{H}} : \mathcal{H} \times \mathcal{H} \to \mathbb{R}/\mathbb{Z}$, where $\mathcal{H} \in \{\mathcal{K}, \mathcal{L}\}$ denotes either $\mathcal{K}$ or $\mathcal{L}$. This pairing $\langle \cdot, \cdot \rangle_{\mathcal{H}}$ on $\mathcal{H}$ is deduced from the pairing $\langle \cdot, \cdot \rangle$ on $\widehat{Z}_{\mathcal{G}}$ and the intersection pairing between 1-cycles on $\mathcal{C}$. To specify the pairing $\langle [C_1], [C_2] \rangle_{\mathcal{H}}$ where $[C_1], [C_2] \in \mathcal{H}$, let us choose two representative K1-cycles $C_1, C_2$ of $[C_1], [C_2]$ respectively. Say the 1-cycles underlying $C_1$ and $C_2$ intersect at $n$ points $p_1, p_2, \cdots, p_n$, and let $t_{1,i}$ and $t_{2,i}$ be the tangent vectors along $C_1$ and $C_2$ respectively at $p_i$. Also let $\alpha_i \in \widehat{Z}_{\mathcal{G}}$ be the element carried by $C_1$ at $p_i$ and $\beta_i \in \widehat{Z}_{\mathcal{G}}$ be the element carried by $C_2$ at $p_i$. Then, we have

$$\left\langle [C_1], [C_2] \right\rangle_{\mathcal{H}} = \sum_i s_i \left\langle \alpha_i, \beta_i \right\rangle, \tag{43}$$

where $s_i = \pm 1$ is defined via

$$t_{1,i} \wedge t_{2,i} = s_i P_i \, \text{vol}, \tag{44}$$

where $P_i > 0$ and vol is a chosen orientation of $\mathcal{C}$. In other words, $s_i = +1$ if the orientation described by $t_{1,i} \wedge t_{2,i}$ matches the orientation of $\mathcal{C}$, and $s_i = -1$ if the orientation described by $t_{1,i} \wedge t_{2,i}$ is opposite to the orientation of $\mathcal{C}$.

Compactifying 6$d$ surface defect along a K2-chain $D$ leads to a 4$d$ local operator living at the end of the 4$d$ line defect corresponding to K1-cycle $C$, where $C = \partial D$. See Figure 3. This local operator can carry flavor charge only if the 2-chain underlying $D$ contains some punctures, i.e. if $D$ is an extended K2-chain. Thus, if one wants to keep track of flavor charges of junction local operators, one should study K1-homology $\mathcal{K}$, but if one does not want to keep track of flavor charges, then one should study K1-homology $\mathcal{L}$. Consequently, the *non-extended* K1-homology $\mathcal{K}$ is not very relevant for the discussion of 1-form symmetry, but would play a crucial role in the discussion of 2-group symmetry in Section 4.4.

On the other hand, the *extended* K1-homology $\mathcal{L}$ was proposed in [31] to be the group, also known as *defect group* [42], formed by 4$d$ line defects modulo screenings and flavor charges, with the mutual non-locality of line defects being captured by the pairing $\langle \cdot, \cdot \rangle_{\mathcal{L}}$ on $\mathcal{L}$ described above. $\mathcal{L}$ was explicitly computed in [31] for all possible Class S theories, but a general invariant description of $\mathcal{L}$ as a K1-homology was not described explicitly. The discussion of this subsection can be viewed as filling that gap. It should be emphasized that this proposal is only valid when all the punctures (twisted or untwisted) are regular punctures. For irregular punctures, we need to add new K1-chains whose underlying 1-chains can end on irregular

punctures, and not all K1-cycles which are boundaries of *extended* K2-chains are trivial if the K2-chain passes over irregular punctures. See Section 5 of [31] and Section 3.2 of [36] for some examples illustrating these modifications.

A genuine (also called 'absolute') 4d $\mathcal{N} = 2$ Class S theory $\mathfrak{T}_\Lambda$ is obtained from the non-genuine, relative Class S theory $\mathfrak{T}$ by choosing a topological boundary condition $\mathcal{B}_\Lambda$ of the associated 5d TQFT and compactifying the 5d TQFT on a segment with one end of the segment occupied by the non-genuine 4d $\mathcal{N} = 2$ theory $\mathfrak{T}$ and the other end of the segment occupied by $\mathcal{B}_\Lambda$. The resulting genuine, absolute 4d $\mathcal{N} = 2$ Class S theory $\mathfrak{T}_\Lambda$ has a group $\Lambda$ of *genuine* line defects modulo screenings and flavor charges, where $\Lambda$, also known as *polarization*, is a maximal subgroup of $\mathcal{L}$ such that

$$\big\langle [C], [D] \big\rangle_{\mathcal{L}} = 0, \qquad \forall\, [C], [D] \in \Lambda. \tag{45}$$

The other line defects lying in $\mathcal{L} - \Lambda$ are non-genuine line defects (modulo screenings) of $\mathfrak{T}_\Lambda$. We can identify the 1-form symmetry group $\mathcal{O}_\Lambda$ of $\mathfrak{T}_\Lambda$ as

$$\mathcal{O}_\Lambda = \widehat{\Lambda}, \tag{46}$$

which is the Pontryagin dual of $\Lambda$.

## 4.3 Global Form of 0-Form Flavor Symmetry Group

Let us now discuss the global form of flavor symmetry group of a Class S theory. It should be noted that we only consider manifest flavor symmetry of the theory encoded in the properties of the regular punctures. The true full flavor symmetry of the theory may be an enhancement of the manifest flavor symmetry. The discussion in this subsection is a review of the contents of [32], to which the reader is referred to for more details. The detailed properties of the punctures required in the following analysis can be found in [43–54].

Let us label the punctures on $\mathcal{C}$ as $\mathcal{P}_i$. If the puncture $\mathcal{P}_i$ lives at the end of an $o$-twist line, then it is characterized by a homomorphism

$$\rho_i: \quad \mathfrak{su}(2) \to \mathfrak{h}_o^\vee, \tag{47}$$

where $\mathfrak{h}_o^\vee$ is the Langlands dual of the subalgebra $\mathfrak{h}_o \subseteq \mathfrak{g}$ left invariant by the action of $o$ on $\mathfrak{g}$. The puncture contributes a flavor symmetry algebra $\mathfrak{f}_i$ to the 4d theory which is given by the commutant of the image

$$\rho_i\big(\mathfrak{su}(2)\big) \subseteq \mathfrak{h}_o^\vee. \tag{48}$$

Let us decompose

$$\mathfrak{f}_i = \bigoplus_\mu \mathfrak{f}_{i,\mu} \oplus \mathfrak{u}(1)^{a_i}, \tag{49}$$

where $\mathfrak{f}_{i,\mu}$ is a non-abelian finite simple Lie algebra and $\mathfrak{u}(1)^{a_i}$ is the abelian part of $\mathfrak{f}_i$. Correspondingly, we associate a group $\widehat{Z}_i$ which is Pontryagin dual of the center $Z_i$ of the group

$$F_i = \prod_\mu F_{i,\mu} \times U(1)^{a_i}, \tag{50}$$

where $F_{i,\mu}$ is the simply connected group with associated Lie algebra $\mathfrak{f}_{i,\mu}$ and $U(1)^{a_i}$ is a group with associated Lie algebra $\mathfrak{u}(1)^{a_i}$. In total, the group of possible flavor center charges is

$$\widehat{Z}_F = \prod_i \widehat{Z}_i. \tag{51}$$

The subgroup $Y_F \subseteq \widehat{Z}_F$ of flavor center charges occupied by genuine local operators receives two different kinds of contributions. First, there is a contribution coming from each puncture $\mathcal{P}_i$:

- If $\mathcal{P}_i$ is a $\mathbb{Z}_2$-twisted puncture for $\mathfrak{g} = \mathfrak{su}(2n+1)$, then the $4d$ theory contains genuine local operators whose flavor center charges lie in a sublattice $Y_{F,i} \subseteq \widehat{Z}_i$ generated by the center charges of irreps contained in a representation $\mathcal{S}_{o,i}^\vee$ of $\mathfrak{f}_i$. $\mathcal{S}_{o,i}^\vee$ is obtained by viewing the *fundamental* representation of $\mathfrak{h}_o^\vee = \mathfrak{sp}(n)$ from the point of view of the subalgebra $\mathfrak{f}_i \subseteq \mathfrak{sp}(n)$ [6].

- If $\mathcal{P}_i$ is not of the above type, then the $4d$ theory contains genuine local operators whose flavor center charges lie in a sublattice $Y_{F,i} \subseteq \widehat{Z}_i$ generated by the center charges of irreps contained in a representation $\mathcal{S}_{o,i}^\vee$ of $\mathfrak{f}_i$ obtained by viewing the *adjoint* representation of $\mathfrak{h}_o^\vee$ from the point of view of the subalgebra $\mathfrak{f}_i \subseteq \mathfrak{h}_o^\vee$.

Combining all these contributions, we find that at least the sublattice

$$Y_F' = \bigoplus_i Y_{F,i} \subseteq \widehat{Z}_F \tag{52}$$

of flavor center charges is realized by genuine local operators of the $4d$ theory.

The second contribution to $Y_F$ arises from K2-cycles, that is from surface defects wrapping the whole $\mathcal{C}$. The idea is that such a surface defect wrapping $\mathcal{C}$ gives rise to a genuine local operator in the $4d$ theory. Since the surface defect passes over each puncture $\mathcal{P}_i$ on $\mathcal{C}$, the corresponding local operator transforms in some representation of each flavor algebra $\mathfrak{f}_i$. The K2-cycles form a group $\mathcal{Y}$ which is either $\mathbb{Z}_m$ or $\mathbb{Z}_2 \times \mathbb{Z}_2$. Choose some generators $g_\alpha$ for $\mathcal{Y}$. For $\mathcal{Y} \simeq \mathbb{Z}_m$, we have a single generator $g_1$, while for $\mathcal{Y} \simeq \mathbb{Z}_2 \times \mathbb{Z}_2$, we have two generators $g_1, g_2$. Each generator $g_\alpha$ provides a contribution $Y_\alpha$ to $Y_F$. In a small neighborhood of a puncture $\mathcal{P}_i$, the K2-cycle $g_\alpha$ carries some element $\beta_i \in \widehat{Z}_\mathcal{G}$ that is left invariant by the outer-automorphism $o$ associated to $\mathcal{P}_i$. Let $R_i$ be an irreducible representation whose charge under $Z_\mathcal{G}$ is $\beta_i$. Then, as discussed in detail in [32], $R_i$ descends to an irreducible representation $R_{o,i}^\vee$ of $\mathfrak{h}_o^\vee$, where $\mathfrak{h}_o^\vee$ is the algebra associated to $\mathcal{P}_i$ as discussed above. Viewing $R_{o,i}^\vee$ from the point of view of the subalgebra $\mathfrak{f}_i \subseteq \mathfrak{h}_o^\vee$ provides us with a (in general reducible) representation $\mathcal{R}_{o,i}^\vee$ of $\mathfrak{f}_i$. Then, $Y_\alpha \subseteq \widehat{Z}_F$ is the sub-lattice generated by center charges of irreps of $\mathfrak{f}$ contained in the representation

$$\mathcal{R} = \bigotimes_i \mathcal{R}_{o,i}^\vee \tag{53}$$

of $\mathfrak{f} = \bigoplus_i \mathfrak{f}_i$.

Combining the two contributions, we have

$$Y_F = \left\langle \bigcup_\alpha Y_\alpha \cup Y_F' \right\rangle, \tag{54}$$

which is the sublattice of $\widehat{Z}_F$ generated by the union of $Y_F'$ and all $Y_\alpha$.

## 4.4 2-Group Symmetries

As discussed in Section 4.2, we should study the non-extended K1-homology $\mathcal{K}$ in order to study 2-group symmetry of Class S theories. The non-extended K1-homology $\mathcal{K}$ is related to the extended K1-homology $\mathcal{L}$ as

$$\mathcal{L} = \mathcal{K}/\mathcal{N}, \tag{55}$$

where $\mathcal{N}$ is the subgroup of elements $[C] \in \mathcal{K}$ such that

$$\left\langle [C], [D] \right\rangle_\mathcal{K} = 0, \qquad \forall\, [D] \in \mathcal{K}. \tag{56}$$

---

[6] We remind the reader that $\mathfrak{h}_o^\vee$ is a subalgebra associated to each puncture $\mathcal{P}_i$ as discussed above.

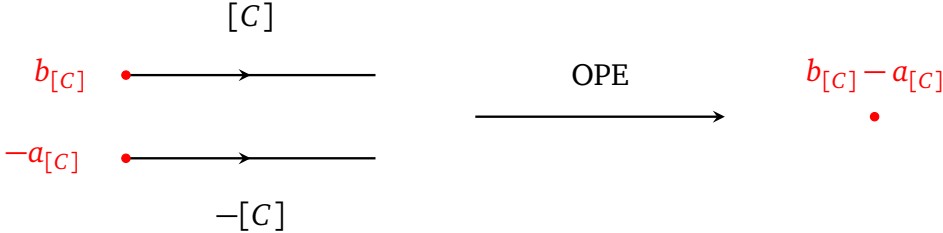

Figure 4: Taking OPE of two local operators living at the ends of two line defects that are inverse of each other leads to a genuine local operator. The flavor center charges are added in this process.

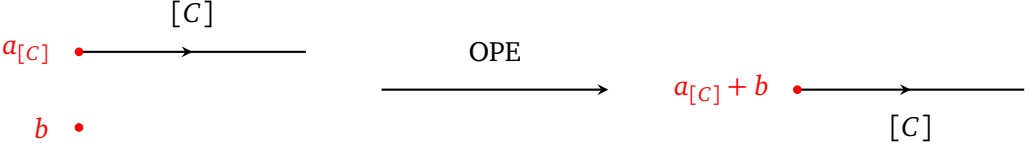

Figure 5: Taking OPE of a genuine local operator with a local operator living at the end of a line defect leads to another local operator living at the end of that line defect. The flavor center charges are added in this process.

That is, the elements in $\mathcal{N}$ have trivial pairing with all elements of $\mathcal{K}$. Physically, this is because an extended K2-chain making an element $[C] \in \mathcal{N}$ trivial gives rise to a local operator that lives at the end of a line operator $L$ in the class $[C]$. Since $L$ ends, it can have no non-locality with other line operators.

Now we want to keep track of the flavor center charges of local operators living at the ends of line defects in $\mathcal{N}$. Let $Y_{[C]} \subseteq \widehat{Z}_F$ be the subset of flavor center charges occupied by local operators living at the end of a line defect $[C] \in \mathcal{N}$. Any two elements $a_{[C]}, b_{[C]} \in Y_{[C]}$ are related as

$$b_{[C]} - a_{[C]} \in Y_F. \tag{57}$$

This follows from taking OPE of line defect $[C]$ with a local operator of charge $b_{[C]}$ living at its end, and a line defect $-[C]$ with a local operator of charge $-a_{[C]}$ at its end. See Figure 4. Moreover, we have

$$a_{[C]} + b \in Y_{[C]}, \tag{58}$$

if $b \in Y_F$ and $a_{[C]} \in Y_{[C]}$. This follows from taking OPE of line defect $[C]$ with a local operator of charge $a_{[C]}$ living at its end, and a genuine local operator of charge $b$. See Figure 5. Thus one only needs to determine a single element $a_{[C]} \in Y_{[C]}$ and then

$$Y_{[C]} = a_{[C]} + Y_F \tag{59}$$

determines the whole $Y_{[C]}$ in terms of $a_{[C]}$.

To determine an element $a_{[C]} \in Y_{[C]}$, pick a K1-cycle $C$ in class $[C]$ and an extended K2-chain $D$ such that $\partial D = C$. Let $\mathcal{D}$ be the set of punctures that lie in the underlying 2-chain associated to $D$. In a local neighborhood of a puncture $i \in \mathcal{D}$, the K2-chain $D$ carries an element $\beta_{D,i} \in \widehat{Z}_G$ that is left invariant by the outer-automorphism $o$ associated to $\mathcal{P}_i$. Let $R_{D,i}$ be an irreducible representation whose charge under $Z_G$ is $\beta_{D,i}$, and let $R^\vee_{o,D,i}$ be the irrep of $\mathfrak{h}^\vee_o$ that descends from the irrep $R_{D,i}$ of $\mathfrak{g}$, where $\mathfrak{h}^\vee_o$ is the algebra associated to $\mathcal{P}_i$ as discussed in Section 4.3. Viewing $R^\vee_{o,D,i}$ from the point of view of the subalgebra $\mathfrak{f}_i \subseteq \mathfrak{h}^\vee_o$ provides us with a (in general reducible) representation $\mathcal{R}^\vee_{o,D,i}$ of $\mathfrak{f}_i$. Then, $a_{[C]}$ can be taken to be the center

charge of any irrep of $\mathfrak{f}$ contained in the representation

$$\mathcal{R}_D = \bigotimes_{i \in \mathcal{D}} \mathcal{R}_{o,D,i}^{\vee} \bigotimes_{i \notin \mathcal{D}} \mathbf{1}_i \tag{60}$$

of $\mathfrak{f} = \bigoplus_i \mathfrak{f}_i$, where $\mathbf{1}_i$ denotes the trivial representation of $\mathfrak{f}_i$.

Now, we can choose $a_{[C]} + a_{[D]}$ as $a_{[C]+[D]}$, from which we see that

$$Y_{[C]} + Y_{[D]} = Y_{[C]+[D]}. \tag{61}$$

Thus, the subset

$$\mathcal{M} := \bigcup_{[C] \in \mathcal{N}} \left( [C], Y_{[C]} \right) \subseteq \mathcal{K} \times \widehat{Z}_F \tag{62}$$

is a *subgroup* of $\mathcal{K} \times \widehat{Z}_F$ which plays the role of 'matter content' for a Class S theory.

We obtain a matrix of exact sequences akin to (28)

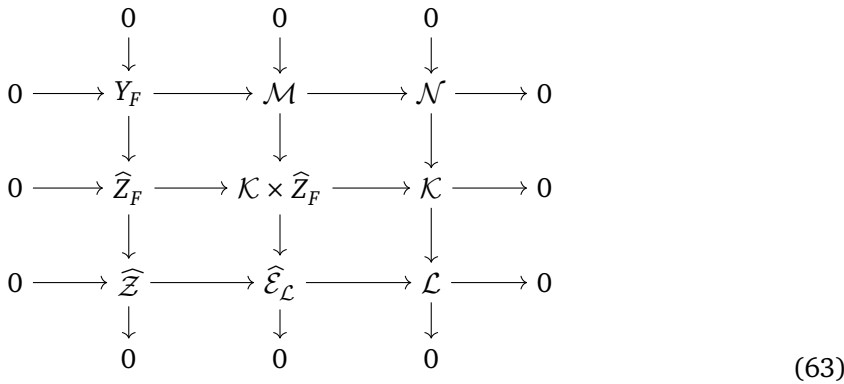

$$\tag{63}$$

where $\widehat{\mathcal{E}}_{\mathcal{L}}$ is the group formed by equivalence classes of line defects plus flavor Wilson lines of the relative 4d $\mathcal{N} = 2$ Class S theory $\mathfrak{T}$.

For a genuine absolute 4d $\mathcal{N} = 2$ Class S theory $\mathfrak{T}_\Lambda$, the above matrix of short exact sequences (63) gives rise to a matrix of short exact sequences when restricted to $\Lambda \subset \mathcal{L}$

$$\begin{array}{ccccccccc}
& & 0 & & 0 & & 0 & & \\
& & \downarrow & & \downarrow & & \downarrow & & \\
0 & \longrightarrow & Y_F & \longrightarrow & \mathcal{M} & \longrightarrow & \mathcal{N} & \longrightarrow & 0 \\
& & \downarrow & & \downarrow & & \downarrow & & \\
0 & \longrightarrow & \widehat{Z}_F & \longrightarrow & \mathcal{K}_\Lambda \times \widehat{Z}_F & \longrightarrow & \mathcal{K}_\Lambda & \longrightarrow & 0 \\
& & \downarrow & & \downarrow & & \downarrow & & \\
0 & \longrightarrow & \widehat{\mathcal{Z}} & \longrightarrow & \widehat{\mathcal{E}}_\Lambda & \longrightarrow & \Lambda & \longrightarrow & 0 \\
& & \downarrow & & \downarrow & & \downarrow & & \\
& & 0 & & 0 & & 0 & &
\end{array} \tag{64}$$

where

$$\mathcal{K}_\Lambda = \pi^{-1}(\Lambda), \tag{65}$$

if $\pi : \mathcal{K} \to \mathcal{L}$ is the projection map associated to modding out $\mathcal{K}$ by $\mathcal{N}$. Consequently, $\widehat{\mathcal{E}}_\Lambda$ is the group formed by equivalence classes of genuine line defects plus flavor Wilson lines of the absolute 4d $\mathcal{N} = 2$ Class S theory $\mathfrak{T}_\Lambda$, which thus has a potential 2-group symmetry governed by the Postnikov class (20) with Bockstein homomorphism associated to the short exact sequence

$$0 \to \mathcal{O}_\Lambda \to \mathcal{E}_\Lambda \to \mathcal{Z} \to 0, \tag{66}$$

which is the Pontryagin dual of the short exact sequence in the bottom-most row of (64). Notice that we can express

$$\widehat{\mathcal{E}}_\Lambda = \widetilde{\pi}^{-1}(\Lambda),\tag{67}$$

if $\widetilde{\pi} : \widehat{\mathcal{E}}_\mathcal{L} \to \mathcal{L}$ is the projection map associated to modding out $\widehat{\mathcal{E}}_\mathcal{L}$ by $\widehat{\mathcal{Z}}$.

# 5 An Illustrative Example

Consider compactifying $\mathfrak{g} = \mathfrak{so}(4n+2)$ $\mathcal{N} = (2,0)$ theory on a sphere with two maximal twisted regular punctures (labeled by $i = 1,2$) and two minimal twisted regular punctures (labeled by $i = 3,4$). See Figure 6. We have $\widehat{Z}_\mathcal{G} = \mathbb{Z}_4$ and the outer-automorphism acts on $\widehat{Z}_\mathcal{G}$ by sending each element to its inverse.

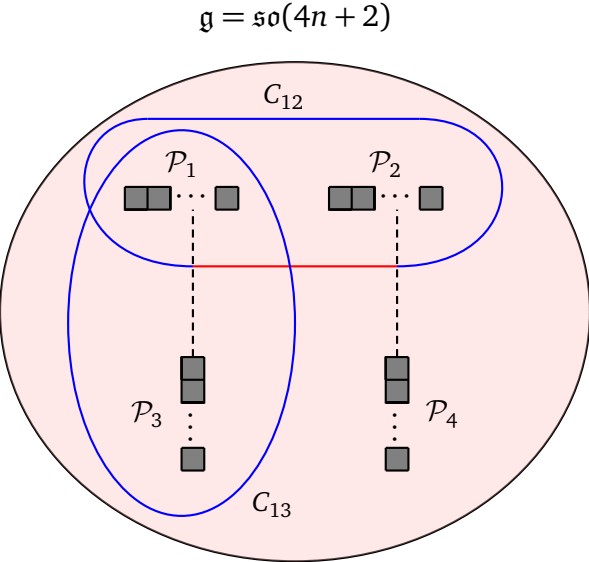

Figure 6: The example considered in the text involves compactification of $6d$ $\mathcal{N} = (2,0)$ theory of $\mathfrak{g} = \mathfrak{so}(4n+2)$ on a sphere carrying four $\mathbb{Z}_2$ twisted regular punctures. The dashed lines display the locus of the $\mathbb{Z}_2$ outer-automorphism twist lines. $\mathcal{P}_1$ and $\mathcal{P}_2$ are maximal punctures and each of them carries an $\mathfrak{sp}(2n)$ flavor algebra. $\mathcal{P}_3$ and $\mathcal{P}_4$ are minimal punctures and neither of them carries a non-trivial flavor algebra. $C_{13}$, which is shown completely in blue, is a cycle encircling $\mathcal{P}_1$ and $\mathcal{P}_3$. $C_{12}$ is a cycle encircling $\mathcal{P}_1$ and $\mathcal{P}_2$, which is divided into two segments, shown in red and blue, separated by outer-automorphism twist lines.

## 5.1 1-Form Symmetry

Let us first compute $\mathcal{K}$ and $\mathcal{N}$. We have cycles $C_i$ surrounding each puncture $i$. Since these cross the outer-automorphism twist line once, only the elements in $\mathbb{Z}_2 \subset \widehat{Z}_\mathcal{G}$ can be placed along them. Thus these cycles lead to non-trivial elements $[C_i] \in \mathcal{K}$ such that $2[C_i] = 0$. In addition to these, we have cycles $C_{13}, C_{12}$ encircling two out of four punctures. See Figure 6. Along $C_{13}$, we can wrap any element of $\widehat{Z}_\mathcal{G}$, leading to a non-trivial element $[C_{13}] \in \mathcal{K}$ such that $4[C_{13}] = 0$. On the other hand, $C_{12}$ crosses twist lines twice, and hence is divided into two sub-segments. Along a sub-segment, we can place any element of $\alpha \in \widehat{Z}_\mathcal{G}$. Then, along the other sub-segment we are forced to place $-\alpha$. Thus this cycle gives rise to a non-trivial element $[C_{12}] \in \mathcal{K}$ such that $4[C_{12}] = 0$. These elements of $\mathcal{K}$ satisfy certain relationships,

which are

$$\sum_i [C_i] = 0,$$

$$[C_1] + [C_2] = 2[C_{12}],$$
$$[C_1] + [C_3] = 2[C_{13}]. \tag{68}$$

Thus, we can choose a basis $[C_1], [C_{12}], [C_{13}]$ for $\mathcal{K}$, implying that

$$\mathcal{K} \simeq \mathbb{Z}_2 \times \mathbb{Z}_4^{(12)} \times \mathbb{Z}_4^{(13)}, \tag{69}$$

where $\mathbb{Z}_4^{(1i)}$ is the $\mathbb{Z}_4$ group generated by $[C_{1i}]$.

The only non-trivial pairing on $\mathcal{K}$ is

$$\langle [C_{12}], [C_{13}] \rangle = \frac{1}{2}. \tag{70}$$

The elements of $\mathcal{K}$ which have zero pairing with all elements of $\mathcal{K}$ are thus $[C_i], 2[C_{12}], 2[C_{13}]$. Thus, we have

$$\mathcal{N} \simeq \mathbb{Z}_2 \times \mathbb{Z}_2^{(12)} \times \mathbb{Z}_2^{(13)} \subset \mathcal{K}, \tag{71}$$

where $\mathbb{Z}_2^{(1i)}$ is the $\mathbb{Z}_2$ subgroup of $\mathbb{Z}_4^{(1i)}$.

Consequently, we have the defect group

$$\mathcal{L} = \mathcal{K}/\mathcal{N} \simeq \mathbb{Z}_2^{[12]} \times \mathbb{Z}_2^{[13]}, \tag{72}$$

where $\mathbb{Z}_2^{[1i]} := \mathbb{Z}_4^{(1i)}/\mathbb{Z}_2^{(1i)}$. It is generated by $[C_{12}], [C_{13}]$ such that $2[C_{12}] = 2[C_{13}] = 0$, and the pairing on $\mathcal{L}$ is given by (70). This reproduces the result of [31].

We have three possible choices of polarization $\Lambda \subset \mathcal{L}$. These are $\Lambda_e = \mathbb{Z}_2^{[13]}$, $\Lambda_m = \mathbb{Z}_2^{[12]}$ and $\Lambda_d = \mathbb{Z}_2^{[23]}$, where $\mathbb{Z}_2^{[23]}$ denotes the diagonal $\mathbb{Z}_2$ subgroup of $\mathbb{Z}_2^{[12]} \times \mathbb{Z}_2^{[13]}$. The respective absolute Class S theories all have 1-form symmetry $\mathcal{O}_\Lambda \simeq \mathbb{Z}_2$.

## 5.2 Global Form of 0-Form Flavor Symmetry Group

Let us compute $Y_F$ now. The punctures $i = 1, 2$ have $\mathfrak{f}_i = \mathfrak{h}_o^\vee = \mathfrak{sp}(2n)$, while the punctures $i = 3, 4$ have $\mathfrak{f}_i = 0$. The total (manifest) flavor algebra is $\mathfrak{f} = \oplus_i \mathfrak{f}_i = \mathfrak{sp}(2n)_1 \oplus \mathfrak{sp}(2n)_2$. To this we associate $F = Sp(2n)_1 \times Sp(2n)_2$.

First let's compute $Y_F'$. Since $\mathfrak{f}_i = \mathfrak{h}_o^\vee$ for $i = 1, 2$, we have $\mathcal{S}_{o,i}^\vee = \mathsf{A}_i$, i.e. the adjoint irrep of $\mathfrak{f}_i = \mathfrak{sp}(2n)_i$. Thus we have $Y_F' = 0$. Now, notice that a K2-cycle must carry the same element of $\widehat{Z}_\mathcal{G}$ at every point of $\mathcal{C}$ and this element must be left invariant by the outer-automorphism, and hence must lie in the $\mathbb{Z}_2$ subgroup of $\widehat{Z}_\mathcal{G} \simeq \mathbb{Z}_4$. Thus $\mathcal{Y} \simeq \mathbb{Z}_2$ and we have a single generator $g_1$. We pick $R_i$ to be vector irrep of $\mathfrak{g} = \mathfrak{so}(4n + 2)$ which descends to $\mathcal{R}_{o,i}^\vee = \mathsf{F}_i$ i.e. the fundamental irrep of $\mathfrak{f}_i = \mathfrak{sp}(2n)_i$ for $i = 1, 2$. Consequently, we have $\bigcup_\alpha Y_\alpha = Y_1 = (\mathbb{Z}_2)_{1,2}$, where $(\mathbb{Z}_2)_{1,2}$ is the diagonal $\mathbb{Z}_2$ subgroup of $\widehat{Z}_F = (\mathbb{Z}_2)_1 \times (\mathbb{Z}_2)_2$ and $(\mathbb{Z}_2)_i$ is the Pontryagin dual of the center of $Sp(2n)_i$.

In total, we have

$$Y_F = (\mathbb{Z}_2)_{1,2}, \tag{73}$$

and

$$\widehat{\mathcal{Z}} = \widehat{Z}_F/Y_F = (\mathbb{Z}_2)_{[1,2]}, \tag{74}$$

where $(\mathbb{Z}_2)_{[1,2]} := \frac{(\mathbb{Z}_2)_1 \times (\mathbb{Z}_2)_2}{(\mathbb{Z}_2)_{1,2}}$. The (manifest) flavor symmetry group of the theory is

$$\mathcal{F} = F/\mathcal{Z} = \frac{Sp(2n)_1 \times Sp(2n)_2}{\mathbb{Z}_2^{\text{diag}}}, \tag{75}$$

where $\mathbb{Z}_2^{\text{diag}}$ is the diagonal subgroup of the $\mathbb{Z}_2 \times \mathbb{Z}_2$ center of $F = Sp(2n)_1 \times Sp(2n)_2$.

### 5.3   2-Group Symmetry

Finally, let us compute $\mathcal{M}$. We choose the K2-chain $D_1$ making $C_1$ trivial to be the one containing the puncture 1 but no other puncture $i \neq 1$. Every point in $D_1$ carries the non-trivial element in the $\mathbb{Z}_2$ subgroup of $\widehat{Z}_{\mathcal{G}} \simeq \mathbb{Z}_4$, which captures flavor center charge of the vector irrep of $\mathfrak{g} = \mathfrak{so}(4n+2)$, which we choose to be $R_{D_1}$. It leads to the representation

$$\mathcal{R}_{D_1} = \mathsf{F}_1 \otimes \mathbf{1}_2 \tag{76}$$

of $\mathfrak{f} = \mathfrak{sp}(2n)_1 \oplus \mathfrak{sp}(2n)_2$, where $\mathsf{F}_i$ denotes the fundamental irrep of $\mathfrak{sp}(2n)_i$ and $\mathbf{1}_i$ denotes the trivial rep of $\mathfrak{sp}(2n)_i$. Thus, we have

$$a_{[C_1]} = (1,0) \in (\mathbb{Z}_2)_1 \times (\mathbb{Z}_2)_2 \,. \tag{77}$$

Similarly, we choose the K2-chain $D_{1i}$ making $2C_{1i}$ trivial to be the one that contains punctures 1 and $i$, where $i \in \{2,3\}$. Again, every point in $D_{1i}$ carries the non-trivial element in the $\mathbb{Z}_2$ subgroup of $\widehat{Z}_{\mathcal{G}} \simeq \mathbb{Z}_4$. We choose $R_{D_{1i}}$ to be the vector irrep of $\mathfrak{g} = \mathfrak{so}(4n+2)$. This leads to

$$\begin{aligned}
\mathcal{R}_{D_{12}} &= \mathsf{F}_1 \otimes \mathsf{F}_2 \,, \\
\mathcal{R}_{D_{13}} &= \mathsf{F}_1 \otimes \mathbf{1}_2 \,,
\end{aligned} \tag{78}$$

from which we obtain

$$\begin{aligned}
a_{2[C_{12}]} &= (1,1) \in (\mathbb{Z}_2)_1 \times (\mathbb{Z}_2)_2 \,, \\
a_{2[C_{13}]} &= (1,0) \in (\mathbb{Z}_2)_1 \times (\mathbb{Z}_2)_2 \,.
\end{aligned} \tag{79}$$

Thus, $\mathcal{M} \simeq \mathbb{Z}_2^4$ is generated by $(1,0,0,1,0), (0,2,0,0,0), (0,0,2,1,0), (0,0,0,1,1) \in \mathcal{K} \times \widehat{Z}_F = \mathbb{Z}_2 \times \mathbb{Z}_4^{(12)} \times \mathbb{Z}_4^{(13)} \times (\mathbb{Z}_2)_1 \times (\mathbb{Z}_2)_2$, from which we compute

$$\widehat{\mathcal{E}}_{\mathcal{L}} \simeq \mathbb{Z}_2^{[12]} \times \left(\mathbb{Z}_4^{[13]}\right)_{[1,2]} \,, \tag{80}$$

where $\left(\mathbb{Z}_4^{[13]}\right)_{[1,2]}$ is a $\mathbb{Z}_4$ such that its $\mathbb{Z}_2$ subgroup is $(\mathbb{Z}_2)_{[1,2]}$ and

$$\left(\mathbb{Z}_4^{[13]}\right)_{[1,2]} / (\mathbb{Z}_2)_{[1,2]} = \mathbb{Z}_2^{[13]} \,. \tag{81}$$

The short exact sequence

$$0 \to \widehat{\mathcal{Z}} \to \widehat{\mathcal{E}}_{\mathcal{L}} \to \mathcal{L} \to 0 \tag{82}$$

in (63) becomes

$$0 \to (\mathbb{Z}_2)_{[1,2]} \to \mathbb{Z}_2^{[12]} \times \left(\mathbb{Z}_4^{[13]}\right)_{[1,2]} \to \mathbb{Z}_2^{[12]} \times \mathbb{Z}_2^{[13]} \to 0 \,. \tag{83}$$

Let us now study 2-group symmetries of various absolute 4d $\mathcal{N} = 2$ absolute Class S theories associated to polarizations $\Lambda_e, \Lambda_m, \Lambda_d$:

- For $\Lambda = \Lambda_m$, we obtain

$$\widehat{\mathcal{E}}_{\Lambda} \simeq \mathbb{Z}_2^{[12]} \times (\mathbb{Z}_2)_{[1,2]} \tag{84}$$

  by using (67). The short exact sequence

$$0 \to \widehat{\mathcal{Z}} \to \widehat{\mathcal{E}}_{\Lambda} \to \Lambda \to 0 \tag{85}$$

  in (64) becomes

$$0 \to (\mathbb{Z}_2)_{[1,2]} \to \mathbb{Z}_2^{[12]} \times (\mathbb{Z}_2)_{[1,2]} \to \mathbb{Z}_2^{[12]} \to 0 \,, \tag{86}$$

  which clearly splits and hence there is no 2-group symmetry.

- For $\Lambda = \Lambda_e$, we have

$$\widehat{\mathcal{E}}_\Lambda \simeq \left(\mathbb{Z}_4^{[13]}\right)_{[1,2]} \tag{87}$$

  by using (67). The short exact sequence (85) becomes

$$0 \to (\mathbb{Z}_2)_{[1,2]} \to \left(\mathbb{Z}_4^{[13]}\right)_{[1,2]} \to \mathbb{Z}_2^{[13]} \to 0\,, \tag{88}$$

  which does not split and hence there is a potential non-trivial 2-group symmetry.

- For $\Lambda = \Lambda_d$, we have

$$\widehat{\mathcal{E}}_\Lambda \simeq \left(\mathbb{Z}_4^{[23]}\right)_{[1,2]}\,, \tag{89}$$

  where $\left(\mathbb{Z}_4^{[23]}\right)$ is the $\mathbb{Z}_4$ subgroup of $\mathbb{Z}_2^{[12]} \times \left(\mathbb{Z}_4^{[13]}\right)_{[1,2]}$ generated by combining the generators of $\mathbb{Z}_2^{[12]}$ and $\left(\mathbb{Z}_4^{[13]}\right)_{[1,2]}$. The short exact sequence (85) becomes

$$0 \to (\mathbb{Z}_2)_{[1,2]} \to \left(\mathbb{Z}_4^{[23]}\right)_{[1,2]} \to \mathbb{Z}_2^{[23]} \to 0\,, \tag{90}$$

  which does not split and hence there is a potential non-trivial 2-group symmetry.

## 5.4 Check Against Gauge Theory Prediction

The above results for polarization $\Lambda = \Lambda_e$ can be verified using a Lagrangian description because the Class S theory under consideration admits a limit under which it becomes a weakly coupled 4d $\mathcal{N} = 2$ gauge theory with gauge algebra $\mathfrak{so}(4n+2)$ and $4n$ hypermultiplets in vector irrep of the gauge algebra. The quiver diagram of the gauge theory can be written as

$$\left[\mathfrak{sp}(2n)_1\right] \xrightarrow{\frac{1}{2}\mathsf{F} \qquad \mathsf{F}} \mathfrak{so}(4n+2) \xrightarrow{\mathsf{F} \qquad \frac{1}{2}\mathsf{F}} \left[\mathfrak{sp}(2n)_2\right]. \tag{91}$$

The $\mathfrak{sp}(2n)_1 \oplus \mathfrak{sp}(2n)_2$ flavor symmetry is realized by splitting the $4n$ hypers into two blocks of $2n$ hypers each. The choice of polarization $\Lambda = \Lambda_e$ corresponds to gauge group being the simply connected group $G = Spin(4n+2)$, in which case we can apply the general analysis of Section 3.

We have $\widehat{Z}_G \simeq \mathbb{Z}_4$ and $\widehat{Z}_F \simeq (\mathbb{Z}_2)_1 \times (\mathbb{Z}_2)_2$. The sublattice $\mathcal{M} \simeq \mathbb{Z}_2 \times \mathbb{Z}_2$ is generated by $(2,1,0),(2,0,1) \in \widehat{Z}_G \times \widehat{Z}_F = \mathbb{Z}_4 \times (\mathbb{Z}_2)_1 \times (\mathbb{Z}_2)_2$. From this, using (27) we compute that $Y_G$ is the $\mathbb{Z}_2$ subgroup of $\widehat{Z}_G = \mathbb{Z}_4$, implying that the theory has a 1-form symmetry which is the Pontryagin dual of

$$\widehat{\mathcal{O}} = \widehat{Z}_G / Y_G \simeq \mathbb{Z}_2\,. \tag{92}$$

Using (26) we compute that $Y_F = (\mathbb{Z}_2)_{1,2}$ is the diagonal $\mathbb{Z}_2$ subgroup of $\widehat{Z}_F = (\mathbb{Z}_2)_1 \times (\mathbb{Z}_2)_2$, which implies that

$$\widehat{\mathcal{Z}} = \widehat{Z}_F / Y_F = (\mathbb{Z}_2)_{[1,2]}\,, \tag{93}$$

and the global form of flavor symmetry group is

$$\mathcal{F} = F/\mathcal{Z} = \frac{Sp(2n)_1 \times Sp(2n)_2}{\mathbb{Z}_2^{\text{diag}}}\,. \tag{94}$$

Moreover, we can compute the short exact sequence

$$0 \to \widehat{\mathcal{Z}} \to \widehat{\mathcal{E}} \to \widehat{\mathcal{O}} \to 0 \tag{95}$$

in (28) to be

$$0 \to \mathbb{Z}_2 \to \mathbb{Z}_4 \to \mathbb{Z}_2 \to 0\,, \tag{96}$$

matching the result (88) obtained above using our general method applicable to any Class S theory. Thus, we have verified the predicted 2-group symmetry using the Lagrangian description.

## Acknowledgements

The author thanks Jihwan Oh, Sakura Schafer-Nameki and especially Yuji Tachikawa for discussions and comments. The author is particularly grateful to Yasunori Lee, Kantaro Ohmori and Yuji Tachikawa for sharing a draft of their soon to appear paper [40] which formed a core inspiration for this work. This work is supported by ERC grants 682608 and 787185 under the European Union's Horizon 2020 programme.

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
