# Peer review of "-Group Symmetries in Class S"

_SciPost Physics, doi:SciPost Phys. 12, 152 (2022)_

## Round 1 · Referee Report · Anonymous (Referee 1) · 2022-2-5

Report

In this paper the author describes a method for identifying 2-group symmetries of 4d $\mathcal{N}=2$ theories of Class S. The method involves starting with a punctured Riemann surface (twists are allowed, but all punctures are required to be regular) and wrapping surface defects of the 6d $(2,0)$ theory on 1- and 2-cycles/chains, giving rise to line defects and local operators respectively in the 4d theory. The local operators can serve as junctions for the line defects, with any two line defects related by such a junction having the same 1-form symmetry charge. In other words, the 1-form symmetry group is given by the Pontryagin dual of the Abelian group whose elements are equivalence class of lines $\{L_i\}/\sim$, with $L_1 \sim L_2$ if there exists a local junction operator connecting $L_1$ and $L_2$.

The key observation is that if the local junction operators are charged under a 0-form flavor symmetry, the 0- and 1-form symmetries can have non-trivial interplay, giving rise to a 2-group. The goal then becomes to identify the flavor charges of the local operators. This is again possible from examination of the Riemann surface—schematically, when the 2-chains giving the local junction operator pass over a puncture, the corresponding 4d local operator will be charged under the flavor symmetry associated with that puncture.

Altogether, this paper is insightful and very clearly written. In addition, the rather technical discussion in Section 4 is complimented with a beautifully explicit example in Section 5. For these reasons, I recommend this paper for publication in SciPost.

Requested changes

One very minor correction: above (2.19), I believe "2-form symmetry” should be “2-group symmetry”. Also, I think that the equality in (2.19) should actually be $\neq$?

---

## Round 1 · Referee Report · Anonymous (Referee 2) · 2022-4-13

Strengths

1) Very clear discussion of generalities of 2-group symmetries

2) A nice algorithm to determine 2-group structure of class ${\cal S }$ theories

Weaknesses

No weaknesses I could find.

Report

In this paper the author studies the 2-group structures in class ${\cal S}$ theories. The discussion does not rely on a Lagrangian description of such theories, which almost always is absent, but rather on geometric understanding of various line operators. The author starts in section 2 discussing and developing a general language and set of tools to understand 2-group structures from properties of line operators of a given theory. In section 3 the methods are applied to gauge theories while in section 4 the machinery developed in previous sections is applied to class S theories. Finally, in section 5 an author discusses an example of a class S theory with a non trivial 2-group which can be also analyzed using a Lagrangian.

This paper is very well written. It is on a very topical subject of understanding higher group and higher form symmetries of general SCFTs. In particular it makes interesting use of geometric constructions of such theories. I think the result of this paper will be interesting to many researchers working in the field. I thus recommend it for publication in SciPost.

---

## Editorial Decision

published